# Brain and Immune System Part II—An Integrative View upon Spatial Orientation, Learning, and Memory Function

**DOI:** 10.3390/ijms262311567

**Published:** 2025-11-28

**Authors:** Volker Schirrmacher

**Affiliations:** Immune-Oncological Center Cologne (IOZK), D-50674 Cologne, Germany; v.schirrmacher@web.de

**Keywords:** dendritic cell, chemokine, engram, neuron, immunotherapy, spacial cells, synapses, T cell receptor, transcriptional memory

## Abstract

The brain and the immune system communicate in many ways and interact directly at neuroimmune interfaces at brain borders, such as hippocampus, choroid plexus, and gateway reflexes. The first part of this review described intercellular communication (synapses, extracellular vesicles, and tunneling nanotubes) during homeostasis and neuroimmunomodulation upon dysfunction. This second part compares spatial orientation, learning, and memory function in both systems. The hippocampus, deep in the medial temporal lobes of the brain, is reported to play a central role in all three functions. Its medial entorhinal cortex contains neuronal spatial cells (place cells, head direction cells, boundary vector cells, and grid cells) that facilitate spatial navigation and allow the construction of cognitive maps. Sensory input (about 100 megabytes per second) via engram neurons and top down and bottom up information processing between the temporal lobes and other lobes of the brain are described to facilitate learning and memory function. Output impulses leave the brain via approximately 1.5 million fibers, which connect to effector organs such as muscles and glands. Spatial orientation in the immune system is described to involve gradients of chemokines, chemokine receptors, and cell adhesion molecules. These facilitate immune cell interactions with other cells and the extracellular matrix, recirculation via lymphatic organs (lymph nodes, thymus, spleen, and bone marrow), and via lymphatic fluid, blood, cerebrospinal fluid, and tissues. Learning in the immune system is summarized to include recognition of exogenous antigens from the outside world as well as endogenous blood-borne antigens, including tumor antigens. This learning process involves cognate interactions through immune synapses and the distinction between self and non-self antigens. Immune education via vaccination helps the process of development of protective immunity. Examples are presented concerning the therapeutic potential of memory T cells, in particular those derived from bone marrow. Like in the brain, memory function in the immune system is described to be facilitated by priming (imprinting), training, clonal cooperation, and an integrated perception of objects. The discussion part highlights evolutionary aspects.

## 1. Introduction

A most important trend in evolution of vertebrate nervous systems is the great elaboration of the size, configuration, and functional capacity of the brain. This encephalization has brought to full fruition several functional capabilities, including fast responses, great capacity for storage of information, and enhanced complexity and flexibility of behavior. In addition, it allowed the formation of associations between past, present, and, at least in humans, future events [1].

In parallel to the brain, the adaptive immunity system developed about 500 million years ago in vertebrates. Mature B and T lymphocytes of the adaptive immunity system express highly diverse antigen-specific receptors. Germline immunoglobulin (Ig) and T cell receptor (TCR) genes are composed of multiple DNA segments that are spatially separate in all cells but are combined in developing lymphocytes [2].

The previous report compared intercellular communication and revealed similarities and differences between the brain and the immune system [2]. In both systems, cognate interactions occur and transfer information via cellular synapses. Homeostasis is maintained by a balance between excitatory and inhibitory protein communities (brain) and between positive and negative signaling receptors (immune system). Electrical signals occur exclusively in the nervous system, while chemical signals are exchanged in both systems. More basic information can be obtained from textbooks about the brain [3,4] and the immune system [5,6].

While vertebrates move through space and time, they require orientation, and they have to learn and memorize the information. The central nervous system (CNS) developed to cope with these challenges. Similarly, to protect the organism against internal dangers, the immune system developed. This required spatial orientation within and between organs and tissues and learning and establishing protective memory. A central immune system (CIS) in bone marrow (BM) seems to cope with these particular functions (see Section 5.4 and Section 7.5).

Apart from visual nerve inputs, there also exist orientation, learning, and memory capacities in different parts of the brain with regard to other nerve inputs, e.g., olfactory or acoustical impressions. The immune system keeps a library of trained and memory cells in different parts of the body from old or recent exposures to infectious agents or other antigenic challenges. The review will describe how the brain and the immune system deal with spatial orientation, learning, and memorizing.

## 2. Brain: Spatial Orientation

### 2.1. Electrical Activity in Neurons

To rapidly transmit signals over large distances, neurons use electrical activity. This is faster than biochemical signaling pathways. In cases of lumbago, for instance, a pain signal is transported in a fraction of a second from the hip via the long myelinized axon of the sciatic nerve to the toe. Electrical activity in brain is measured by electroencephalography (EEG). This allows us to study different waves or rhythms in the normal and abnormal brain [4].

In most resting neurons, the membrane potential (Vm) is between −60 and −75 millivolts. The neuron is polarized: the inside of the plasma membrane (PM) has a negative charge compared to the outside, due to uneven distribution of ions (e.g., Na^+^, K^+^, Cl^−^, Ca^2+^, HCO3^−^). Since ions carry charges and therefore electrical currents, ion channels provide the main pathway for electrical signaling in neurons. Electrical gradients across PMs are due to ongoing activity of ion transporters (large membrane-spanning proteins with a pore in the center, e.g., Na^+^K^+^ ATPase) and pumps and to the fact that the membrane is semipermeable to certain ions. Most ion channels are not constitutively open but are gated by external stimuli such as changes in Vm. Neuronal electrical activity, manifested in action potentials, is based on intermittent current flow, mostly through voltage-gated ion channels, key elements of the nervous system. Neuronal networks are organized across micro-, meso-, and macro-scale principles and distances [2,4,7].

### 2.2. Neuronal Spatial Cells

Neuroscientific research over the past four decades has revealed that cells in the hippocampal formation of the brain provide an exquisitely detailed representation of an animal’s location and heading [8,9]. The hippocampus is a phylogenetically ancient and well-preserved structure within the limbic system (LS), which in humans is found deep in the medial temporal lobes [10]. There is now a growing understanding of the mechanisms of spatial cognition by neurons in mammals, including humans. The major categories of spatial cells are, in terms of discovery, place cells (1971), head direction cells (1990), boundary vector cells (2000), and grid cells (2005) [11].

Grid cells are a key component of the medial entorhinal cortex (MEC). This is part of a neuronal system for mapping the position of an individual within a physical environment [8,9]. Grid cells fire in a characteristic hexagonal pattern. They are organized in modules that form a population code for the animal’s position. The joint activity of grid cells from an individual module resides on a toroidal manifold [11]. The beautiful periodicity of grid fields has attracted widespread attention.

Head direction (HD) cells are another type of neuron involved in spatial orientation. Each HD cell has a preferred direction corresponding to a compass direction [8]. Phase precession relative to turning angle was reported in theta-modulated HD cells [12]. Also in HD cells, an intrinsic attractor manifold and population dynamics were suggested to form a canonical cognitive circuit across waking and sleep [13].

Boundary cells have firing fields very close to the edges of the environment; they encode short boundary vectors [8].

Place cells are principal cells in the hippocampus proper and dentate gyrus (DG). These cells fire at a low rate throughout most of the environment, but each cell shows increased firing when the animal is within a circumscribed place field [8,9].

### 2.3. Spatial Navigation and Cognitive Maps

Hippocampal cells fire selectively in specific and restricted locations (place fields) as rodents move through open environments [11]. The hippocampal brain area CA3 contains place cells whose location-selective firing fields implement maps supporting spatial memory [14]. Place cells were shown to emerge in networks (place fields) trained to remember temporally continuous sensory episodes [14]. As animals navigate, grid cells in MEC are activated when the animal passes through multiple locations (firing fields) arranged in a hexagonal lattice [15].

The hippocampal trisynaptic circuit is a relay of synaptic transmission that is made up of three major cell groups: granule cells in the DG, pyramidal neurons in CA3, and pyramidal neurons in CA1. MEC sends signals to the DG via the perforant path. DG sends signals to CA3 via mossy fibers. CA3 sends signals to CA1 via Schaffer collaterals. CA1 sends signals back to the MEC. This circuit is a key part of spatial orientation, learning, and memory [14].

Circuits within the hippocampus are remarkably plastic [9]. This plasticity is mediated in part through changes in synaptic strength and revealed by long-term potentiation (LTP) and long-term depression (LTD) [9]. Cognitive maps and learning have been improved in mice with genetically modified N-methyl-D-aspartate (NMDA) receptors that enhance LTP induction [9]. This glutamate receptor is the human brain’s primary excitatory neurotransmitter.

In the mammalian brain there exist neural codes for 2D and 3D space [11]. Layer II of the MEC comprises grid cells that support spatial navigation [11]. Four types of excitatory neurons could be distinguished in MEC that exhibit cell-type-specific local excitatory and inhibitory connectivity [11]. Local and distal input was reported to control excitation in layer II of the MEC [11].

Spatial frameworks in the brain may be tied to a particular body part, object, or action. For example, neurons in primary visual cortex might respond to a stimulus in a particular part of the visual field, a neuron in primary somatosensory cortex might respond to a tactile stimulus of a particular body part, and the firing of a motor neuron might help to direct limb movements in a specific direction [8]. This type of representation is crucial for behaviors such as catching a ball or picking a fruit from a tree [8].

### 2.4. Spatial Orientation via Acustic and Magnetic Signals

Recordings were made from MEC cells in freely flying bats receiving signals from an echo sounder. The study identified several classes of spatial neurons, including 3D border cells, 3D HD cells, and grid cells with multiple 3D firing fields. While 2D grid cells formed a global lattice, the 3D grid cells exhibited a locally ordered metric for space [15].

A light-dependent magnetic compass exists in terrestrial organisms to help directional and spatial perception [16]. Sensitivity to the Earth’s magnetic field is mediated by at least two different magnetoreception mechanisms. One involves biogenic ferromagnetic crystals (e.g., magnetite), the other involves a photo-induced biochemical reaction that forms long-lasting spin-coordinated radical pair intermediates. In some vertebrate groups (e.g., amphibians and birds), both mechanisms are present [16].

The light-dependent magnetic compass sense of night-migratory songbirds could be disrupted by broadband 75–85 MHz radiofrequency [17]. This finding supports a quantum mechanical radical-pair mechanism of magnetoreception as observed for isolated cytochrome 4, a protein found in birds’ retinas [17]. Support for the existence of magnetic receptors comes from studies of vertebrate retinas, beaks, noses, and inner ears [18]. Neural correlates of a magnetic sense have been reported for the pigeon’s brainstem. There, single cells encode magnetic field direction, intensity, and polarity. This is a prerequisite to derive an internal model representing directional heading (navigation) and geosurface location (orientation) [18]. Recent progress in neuronal circuits and the magnetic sense of birds and rodents has been reviewed [19]. The review highlights the role of the vestibular and trigeminal systems as well as that of the hippocampus. It is speculated (i) that magnetic circuits share anatomical motifs with other senses culminating in the formation of spatial maps in telencephalic areas of the brain and (ii) that spatial cells exist that encode defined components of the Earth’s magnetic field [19].

### 2.5. Spatial Orientation via Olfactory Signals

Olfaction, an evolutionarily ancient sense, is key to learning where to find food, shelter, mates, and important landmarks. In mammals, the anterior olfactory nucleus, piriform cortex, entorhinal cortex, and hippocampus each represent different aspects of olfactory and spatial information [20]. Odor cues are recognized as landmarks by place cells in CA1, thereby improving navigation [21]. Spatial orientation via olfactory signals seems of importance for animals living on the earth, like snakes, or, in the absence of light, within the earth, like moles, or in the deep sea.

### 2.6. Localization of Major Functions of the Cerebrum and the Cerebellum

Spatial orientation in the outside world has its correspondence in the brain, its cerebrum, and cerebellum and its different lobes [1].

The development of the cerebellum is directly correlated with a vertebrate’s mode of locomotion, agility of limb movement, and balance. The unconscious mind—all of the brain except the cerebrum (cerebral cortex)—governs vital functions: respiration, blood pressure, heart rate, hunger, thirst, temperature balance, salt balance, sexual drive, and basic emotions [1].

The cerebrum contains discrete motor (frontal lobe) and sensory (parietal lobe) areas involved in functions of the leg, trunk, arm, hand, face, and tongue. The frontal lobe is also involved in speech and smell. The occipital lobe is focused on vision, while the temporal lobe is focused on hearing. The cerebrum also contains “silent” regions, called associated areas, concerned with memory, judgment, reasoning, and other integrative functions [1].

The brain is also a complex endocrine gland that regulates and receives feedback from the body’s endocrine system [1].

The right and left hemispheres of the cerebrum are bridged through the corpus callosum. In humans, the two hemispheres are specialized for entirely different functions: the left side for language development, mathematical and learning capabilities, and the right side for spatial, musical, artistic, intuitive, and perceptual activities. Since the differences in brain symmetry and function exist at birth, they appear to be inborn rather than the result of developmental or environmental effects [1].

### 2.7. Summary

In summary, there is a growing understanding of the mechanisms in the brain involved in spatial orientation. Deep in the hippocampus reside different types of spatial neurons, such as place cells, head direction cells, boundary cells, and grid cells. Neural codes and maps exist for 2D and 3D space. Neurons in other parts of the brain are tied to a particular body part, object, or action. A light-dependent magnetic compass exists in terrestrial organisms to help directional and spatial perception. Neural correlates of a magnetic sense have been reported for the pigeon’s brainstem. Olfactory signals help animals to find food, shelter, mates, and important landmarks on earth and in the deep sea.

## 3. Brain: Learning

### 3.1. Imprinting

K Lorenz, Nobel laureate in physiology from 1973, founded the field of ethology. This research is concerned with the organization and elicitation of individual and social behavior patterns in animals. Of particular relevance to the topic of this article is Lorenz’ work on imprinting in geese. It is the process by which a young animal forms a strong attachment to its first moving object. Input signals are received via sensory organs such as eye, nose, tongue, ear, and skin. The term “imprinting” was coined to refer to cases of “programmed learning” during a critical period. Lorenz found that this period of particular sensitivity to experience in geese is limited to a few hours soon after birth and that once imprinting occurs, it is irreversible [22].

The relatively new field of developmental cognitive neuroscience tries to elucidate such behavior at a cellular and even molecular level. As one example, it was found that inhibitory processes regulate the time course of the critical period [23]. The maturation of gamma-aminobutyric acid (GABA) ergic inhibition sets the threshold for the start of the critical period for experience-dependent plasticity by enabling visual cortical neurons to detect differences in the activity between competing retinal inputs [23].

### 3.2. Input and Output

According to general sensory physiology, a stimulus can be regarded as a physical or chemical energy pattern that impinges on a primary receptor of a sensory organ and is transduced into a neural signal [4]. Action potentials elicited by light stimuli in the receptor of the eye of a horseshoe crab are directly dependent on the light intensity. For strong intensities there are about 30 impulses per second. Reducing the light intensity to one ten-thousandth of the initial value lowers the impulse frequency to 3 impulses per second [4].

A stimulus (or sensory input) from the eye to the brain is transferred by thick fiber bundles with millions of nerve fibers. By adding fiber bundles from the other sense organs, one reaches a number of 2.5 million fibers for information transfer to the brain. The amount of information that reaches our brain is about 100 megabytes per second.

After information processing by interneurons, output impulses leave the brain via about 1.5 million fibers, which connect to effector organs such as muscles and glands. Information processing includes information exchange between the prefrontal cortex and temporal cortex (top down) and vice versa (bottom up) [3,4].

A simplified model may serve as example: In the retina of the eye of a frog are three sensory neurons (A, B, C) that connect to three output neurons (1, 2, 3). Depending on the pattern that the sensory neurons observe in the outside world, the patterns of activation of the output neurons are different. If the eye sees a stork, the output neurons activate muscles of the thigh to jump away. If the eye sees a fly, the output neurons activate muscles of the tongue for eating and glands for secretion of gastric juice. If the eye sees blue sky, the input signal is low. In this case there is no need to react, and energy can be saved. This is called habituation. Thus, the strength of the synaptic signal appears decisive for reactions to representations [1,3,4].

The central executive network (CEN) includes regions such as the dorsolateral prefrontal cortex, anterior cingulate cortex and inferior parietal lobe. It is central to managing the demands of cognitively challenging motor tasks. A recent review highlights the role of physical exercise to strengthening the CEN and promoting brain health, offering a strategy to improve cognitive resilience and emotional well-being across the lifespan [24].

### 3.3. Forms of Learning

(i) Associative learning. Associative learning comprises the combination of two events or stimuli. The brain learns the correlation between two events (classical conditioning) or between its own action and an event (operant conditioning). The Pavlovian form of learning associates a conditional stimulus (CS) (e.g., sound of a bell) with an unconditional stimulus (US) (e.g., food) resulting in a salivary response.

Another example is the autonomic and behavioral response following auditory fear conditioning as studied in mice: Auditory inputs (CS) via auditory thalamus and auditory cortex, are associated with inputs (US, footshock) from somatosensory thalamus and somatosensory cortex to converge onto single neurons within the lateral nucleus of the amygdala (LA). LA then projects to the central nucleus of the amygdala (CE). Projections from CE to the brainstem and hypothalamus regulate the fear response, including freezing, increased blood pressure and rate and endocrine responses [4]. Molecular mechanisms underlying fear acquisition and consolidation have been described [4].

(ii) Nonassociative learning. The simplest form of learning is habituation. In this situation the animal learns about the irrelevance of a stimulus.

(iii) Complex learning. Imprinting (see above), observational learning, and priming are forms of complex learning. Imprinting is a form of prepared learning. The neuronal circuits involved are only plastic during a very limited period of time. Additional examples of the Lorenz studies are vocal learning in birds or the learning of the mother language in humans. Similarly to human language, songbird singing is a complex motor skill learning behavior that is regulated by an interconnected network of neuronal nuclei in the brain. As an important neurotransmitter, dopamine plays a role in the learning and maintenance of songbirds’ singing behavior [25]. Observational learning can be defined as learning by imitation. In priming, the first stimulus activates part of the neuronal network in a specific brain region just before carrying out an action. The network is already activated when the second stimulus is encountered. This improves performance and is part of the implicit (procedural) memory system [3,4].

The power of the neuronal impulse, expressed by the frequency of the action potential, is transmitted chemically via the concentration of neurotransmitter. This chemical transformation process is associated with the opening of calcium channels in pre-synaptic neurons and of sodium channels in the post-synaptic neurons [2,3,4].

### 3.4. Model Systems

The most widely used organisms for studying the cellular basis of learning and memory processes are mice, the fruit fly drosophila, the sea slug aplysia, and honeybees. The famous dance language and orientation of bees has been described by K Frisch [26]. Pheromone communication in social insects capable of forming superorganisms (ants, wasps, bees, termites) has been summarized in a special volume of Springer Press [27].

## 4. Brain: Memory Function

This chapter deals with four types of memory: episodic memory (Section 4.2), working memory (Section 4.3), semantic memory (Section 4.4), and spatial/time memory (Section 4.5). This is embedded in basic information about developmental and functional aspects (Section 4.1) and about brain oscillations and resonance (Section 4.6).

### 4.1. Memory Development and Functional Aspects

Many mammals are born in a very immature state and develop their rich repertoire of behavioral and cognitive functions postnatally [28]. This development goes in parallel with changes in the anatomical and functional organization of cortical structures. All cortices start with uncorrelated activity in uncoupled single neurons. Shortly after birth, cortical networks develop weakly coordinated multineuronal discharges, which have been termed synchronous plateau assemblies (SPAs). These patterns rely mostly on electrical coupling by gap junctions. The subsequent increase in number and maturation of chemical synapses leads to the generation of large-scale coherent discharges. These patterns have been termed giant depolarizing potentials (GDPs) for predominantly GABA-induced events or early network oscillations (ENOs) for mostly glutamatergic bursts, respectively [28]. During the third to fourth postnatal week, cortical areas reach their final activity patterns with distinct network oscillations and highly specific neuronal discharge sequences, which support adult behavior [28].

Neuronal migration is a key process in the developing and adult brain. Major signaling pathways in neuronal stem cells (NSC) and neuroblasts migrating from the subventricular zone to the final destination, e.g., the olfactory bulb, have been analyzed and described [29].

A simple neuron model, the calcitron, has been proposed [30]. It is a calcium control hypothesis that implements many learning rules that affect memory later. The model is built upon the following four sources of [Ca^2+^]: (i) local (following the activation of an excitatory synapse and confined to that synapse), (ii) heterosynaptic (resulting from the activity of other synapses), (iii) postsynaptic spike-dependent, and (iv) supervisor-dependent. A wide range of learning and plasticity protocols could be reproduced by modulating the plasticity thresholds and calcium influx from each calcium source [30].

Synaptic connectivity in groups of pyramidal neurons in the neocortex was investigated and revealed that synaptic weights leading to activation function follow very closely the number of connections in a group of neurons [31]. Synaptic weight is a numerical value and reflects the strength of synaptic connections between two Ranvier nodes. Activation was achieved after only 20% of possible connections were formed between neurons in a group. The elementary neuronal groups were described as Lego-like building blocks of perception that can be assembled into higher-order constructs to achieve acquired memory [31].

Ancestral brainstem noradrenergic neurons in the locus coeruleus (LC) have been conserved across vertebrates. Evolution has woven the LC into wide-ranging neural circuits. These influence autonomic systems, the stress response, nociception, sleep, and high-level cognition [32].

Deep learning is an advanced artificial intelligence (AI) technology. It offers powerful capabilities for data analysis and pattern recognition. It could help bioinformatics methods in handling complex data patterns such as those derived from genomics. A recent review focuses on the following four major deep learning models: convolutional neural network, recurrent neural network, long short-term memory, and generative adversarial network [33].

A quantum-like model of neuronal network dynamics has been proposed [34]. The model favors information processing of neural activity. Thirteen three-neuron motif classes with a quantum-like nature seem to have a bearing on the observed structural integrity [34]. Organization principles of single-neuron projectomes of mouse hippocampus (n = 10,100) revealed 43 projectome subtypes [35]. Such studies may provide a structural basis for understanding whole-brain spatial organization of hippocampus projectomes [35].

While memories may persist for the lifetime of an organism, the proteins and lipids that make up synapses undergo constant turnover with lifetimes from minutes to days [36]. The molecular basis for memory maintenance may rely on a subset of long-lived proteins. Such long-lived synaptic proteins were indeed identified by proteomic analysis of synaptosome protein turnover with half-lives of several months or longer [36].

### 4.2. Engram Neurons, Coding, and Top-Down Formation of Episodic Memory

Episodic memory is the ability to recall specific events and experiences, a cornerstone of human cognition. Human episodic memory includes integrated *what-where-when*, source memory, free recall temporal binding, and threshold retrieval dynamics [37].

According to a hypothetical biological concept, engram neurons are involved in encoding (shaped by intrinsic properties), consolidation (participation in synaptic- and systems-level consolidation), retrieval (neural reactivation, memory recall), and forgetting (neurobiological mechanisms) [37]. Intrinsic neuronal excitability—the propensity of a neuron to fire an action potential in response to an input—can be a key determinant of participation in memory. It mediates the formation of associated neuronal ensembles (engrams) and may prime for later synaptic plasticity. It outcompetes less excitable counterparts. Converging evidence for this competition-based rule has been obtained across an array of memory assays and neural regions. This rule therefore seems to be a generalizable feature of learning, and thus key for understanding the mechanisms underlying neuronal memory encoding [38]. Memory engrams provide stability and flexibility [39].

The activity of spatially and object-tuned cells contributing to episodic memory is regulated top down by commissural input from MEC [40]. Commissural input derives from extensive interhemispheric projections that connect homotypic brain regions [41]. Coding properties of spatially tuned neurons from MEC are differentially controlled by two septal-entorhinal GABAergic projections expressing either parvalbumin or calbindin [41].

Dual-site in vivo recording in freely behaving mice revealed that hippocampal dorsal CA 1 (dCA1) and basolateral amygdala (BLA) utilize distinct coding strategies for novel experiences [42]. Machine learning decoding suggested that dCA1 population spikes predicted BLA assembly firing rate. Such “many-to-one weighted mapping” in hippocampus-amygdala network emerges to underlie memory formation [42]. The CA1 output region of the hippocampus plays an essential role in the retrieval of episodic memories [43]. Photostimulation of septal GABAergic terminals in CA1 selectively inhibited interneurons. The medial septum (MS) to CA1 connection was found to be recruited during recall of a contextual fear memory. This is an important mechanism in gating contextual fear behavior [43].

Gap junctions containing connexin 36 electrically couple interneurons in many brain regions and synchronize their activity. Such junctions were shown to be required for normal spatial coding, cognition, and short-term spatial memory [44].

### 4.3. Working Memory

The dorsolateral prefrontal cortex (dlPFC) is implicated in working memory (WM) in addition to emotional and cognitive processing [45]. The maintenance of complex visual scenes in WM may require activation of WM manipulation circuits [45]. Dynamic layer-specific processing (functional laminar circuitry) was suggested in the dlPFC during WM [46].

Frontoparietal network topology has been proposed as a neural marker for musical perceptual abilities [47]. The linkage between functional networks and musical abilities was mediated by WM processes, whereas structural networks influenced these abilities through sensory integration [47].

Recent memory consolidation studies emphasize synaptic weight dynamics (correlations between presynaptic and postsynaptic firing rates) [48] and spatio-temporal mechanisms [49]. Systems consolidation is a common feature of learning and memory systems, in which a long-term memory initially stored in one brain region becomes persistently stored in another region [48]. The dynamics were studied in an early-learning and in a late-learning brain area [48]. Recall and reconsolidation in reward-related memory revealed the central amygdala as critical for linking appetitive emotional states with spatial contexts [49].

### 4.4. Post-Encoding Period, Bottom-Up Transfer to Cortex, and Formation of Semantic Memory

Many studies have shown that memories are encoded in sparse neural ensembles distributed across the brain. During the post-encoding period, often during sleep, many of the cells that were active during encoding are reactivated, supporting bottom-up consolidation of this memory. During memory recall, many of the same cells are reactivated. Recent studies suggest that these stable memories and their representations are much more dynamic and flexible than previously thought [39].

A biologically plausible systems-level theory of learning and memory in cortex has proposed four basic kinds of tasks, each requiring some circuit modification: hierarchical memory formation, pairwise association, supervised memorization, and inductive learning of threshold functions [50]. Memories are believed to be stored in distributed neuronal assemblies through activity-induced changes in synaptic and intrinsic properties [50]. A simplified network model incorporating multiple plasticity processes suggests that linked memories share synaptic clusters within the dendrites of overlapping populations of neurons. The model generates numerous experimentally testable predictions regarding the cellular and subcellular properties of memory engrams as well as their spatiotemporal interactions [50].

The interplay between the hippocampus and prefrontal cortex (PFC) is fundamental to spatial cognition. By complementing hippocampal place coding, prefrontal representations provide more abstract and hierarchically organized memories suitable for decision making [51]. Spatial learning and action planning have been described in a PFC network model [51].

Conserved principles also guide olfaction [52]. There are striking similarities between species in the organization of the olfactory pathway, from the nature of the odorant receptor proteins, via perireceptor processes, to the organization of the olfactory CNS, and finally to odor-guided behavior and memory. These common features span a phylogenetically broad array of animals, implying that there is an optimal solution to the problem of detecting and discriminating odors [52]. The olfactory bulb, with its glomeruli, acts as the first processing and refining site of olfactory information. The lateral entorhinal cortex (LEC) relays this olfactory information further to the hippocampus for interpretation. Spatially segregated feedforward and feedback neurons support differential odor processing in the LEC [53].

In Parkinson’s disease (PD), memory retention could be modified by low-frequency deep brain stimulation in non-rapid eye movement (REM) sleep [54]. During non-REM sleep, neuronal activity involves up-down states (UDS), which are synchronous cortical events. Evidence was provided that layer 3 (L3) MEC is crucial in the generation and maintenance of UDS in the MEC. Deep layer L5b MEC was suggested to act as a coincidence detector during information transfer between the hippocampus and the cortex. It thereby plays an important role in memory encoding and consolidation [55].

During sleep, targeted memory reactivation (TMR) enhances memory consolidation by presenting reminder cues. These can be sounds associated with memory. TMR can be applied to post-traumatic stress disorder (PTSD), a psychiatric dysfunction. TMR led to stimulus-locked increases in slow oscillation and spindle dynamics. These increases correlated positively with PTSD symptom reduction [56].

PD and PTSD are two examples of dysfunction of homeostasis between the brain and the immune system. Part I of this review mentions migraine, multiple sclerosis (MS), and brain cancer as three further examples of dysfunction. It also presents neuroimmunomodulatory procedures to counteract such diseases.

### 4.5. Spatial and Time Memory

Path integration (PI) is a highly conserved navigation strategy. To demonstrate that grid cells are involved in PI, grid cell activity was selectively disrupted. Impaired PI was seen in mice without grid cell firing because of lacking GluA1-containing alpha-amino-3-hydroxy-5-methyl-4-isoxazole propionic acid (AMPA) receptors [57] or NMDA glutamate receptors [58].

Selective ablation of NMDA receptors on parvalbumin-positive hippocampal interneurons was reported to impair hippocampal synchrony, spatial representations, and WM in NR1 (PVCreknockout) mice [59]. These mice exhibited impaired spatial working as well as impaired spatial short- and long-term recognition memory but showed no deficits in open field exploratory activity and spatial reference learning [59]. A major cortical input area to the hippocampus is the LEC. This is crucial for associative object-place-context memories [60].

The DG receives substantial input from the homologous brain area of the contralateral hemisphere. Recently it was demonstrated that inhibitory long-range projections connecting DGs in the two hemispheres support spatial and contextual memory [61]. Optogenetic silencing of somatostatin-expressing contralateral DG projections during spatial memory encoding resulted in compromised DG-dependent memory [61].

In the rodent hippocampus exists a sequence of spontaneous activities that are precisely timed: early sharp waves progress to theta and gamma oscillations, place and grid cell firing, and sharp wave-ripples that must occur for spatial memory to develop [62,63].

As in radio communication systems, brain can be regarded as network of dynamic, adaptive transceivers that broadcast and selectively receive multiplexed temporally patterned pulse signals [64]. Various forms of multiplexing neural signals have been described: time-division, frequency-division, code-division, oscillatory phase, synchronized channels, oscillatory hierarchies, and polychronous ensembles [62,63].

### 4.6. Oscillations and Resonance

Neurons can encode information via changes in their firing rates. Spatial orientation and spatial memory are embedded in electromagnetic field oscillations and resonance effects produced by the brain [64]. Rhythmic neuronal network activity underlies brain oscillations [64]. Alpha-band (about 10 Hertz (Hz)) oscillations are the dominant oscillations in the human brain [65]. They have an inhibitory function and can be associated with attention and controlled access to stored information [66]. Beta rhythms (13–35 Hz) are the most distributed cortical brain rhythms [66]. They serve multiple domains of human ability: motor control, cognition, memory, and emotion [66]. Results from an orientation detection task revealed alpha and beta band correlates of haptic perceptual grouping [67,68].

Gamma-band oscillations are implicated in memory processes, such as WM, attention, and associative memory. The frequency of 40 Hz is important and can be increased in amplitude by meditation and neurostimulation. Gamma oscillations in the primary somatosensory cortex (S1) in humans were found to correlate with subjective pain perception. They recruit prefrontal and descending serotonergic pathways in aversion and nociception [69]. Sensory nociceptors produce a signal that travels along a chain of nerve fibers via the spinal cord to the brain. The study describes a mechanistic framework for modulation of pain by specific activity patterns in the S1 cortex [69]. A coupled-oscillator model of olfactory bulb gamma oscillations has been described. Cellular oscillator architecture permits stable and replicable ensemble responses not only to odors but also to other sensory stimuli arising from learning-dependent synaptic plasticity [70].

Low-threshold spiking interneurons have a delta-band peak at 8 Hz, while fast-spiking interneurons fire in the gamma frequency band (20–80 Hz) [71]. These distinct neocortical oscillations, driven by cell-type selective optogenetic drive, appear as separable resonant circuits [71]. The General Resonance Theory’s principle states that the combination of micro- to macro-consciousness in coupled field systems is a function of the slowest common denominator frequency of resonance [64].

Theta-band oscillations (4–7 Hz) modulate brain activity during challenging balance tasks and reflect postural stability monitoring [72]. Adaptive Resonance Theory tries to explain how a brain learns to consciously attend to and recognize a changing world [73]. It includes surface, and boundary attention, gamma and beta oscillations, learning of entorhinal grid cells and hippocampal place cells [73]. Hippocampal network oscillations are coordinated and controlled by GABAergic interneurons [74]. Theta rhythm is, in rats, a large-amplitude oscillation. Theta phase precession is a phenomenon that provides insights into how multiple processing elements in the hippocampal formation interact. It is believed to facilitate rapid learning, navigation, and looking ahead. As rodents turn their heads, grid and place cells fire at progressively earlier phases as head direction sweeps over their preferred tuning direction [12].

Neural oscillations have been reported to play a key role in the appearance of grid cells’ toroidal topology [75].

### 4.7. Summary

In summary, this paragraph reports brain memory studies of the last twenty years, the majority being brand new (2023–2025). An important concept is that of engram neurons that are involved in encoding, consolidation, retrieval, and forgetting. Shortly after birth, cortical networks develop weakly coordinated multineuronal discharges that rely mostly on electrical coupling by gap junctions. This is followed by an increase in the number and maturation of chemical synapses. Large-scale coherent discharges have been termed giant GDPs for GABA-induced events or ENOs for mostly glutamatergic bursts. Systems consolidation is a common feature of learning and memory systems. The dynamics were studied in an early-learning and in a late-learning brain area. Important brain areas of interplay are the hippocampus, the amygdala, and the prefrontal cortex.

## 5. Immune System: Spatial Orientation

The normal functioning of the immune system gives rise to immunity, meaning to be exempt from or protected against a devastating disease. The term “immune” originated in the 1500s, before the causes of disease were understood [6].

### 5.1. Spatial Orientation via Chemokine Gradients

Spatial orientation by the immune system differs from that of the brain in that the mobile cells have to travel through fluids (blood, lymph) and tissues. They also have to cross barriers such as blood vessel endothelium, mucous membranes (e.g., lung and intestine), or the blood–brain barrier to fulfill their job of immune surveillance [5,6]. Chemokines and corresponding receptors allow spatial orientation within the immune system.

Chemokines are chemotactic cytokines. They draw leukocytes toward the precise site of injury or infection in a process called chemotaxis [6]. As chemokines diffuse into the tissue surrounding the inflammatory site, a concentration gradient is created with the highest chemokine concentration at the inflammatory site and steadily decreasing concentrations with increasing distance from this area. Innate and adaptive leukocytes express chemokine receptors that cause these cells to migrate up the gradient within the tissue towards the target with the highest chemokine concentration.

Chemokines are a large family of small secreted proteins that signal through cell surface G protein-coupled heptahelical chemokine receptors. The known 43 chemokines belong to four subgroups, i.e., CCL, CXCL, CX3CL, and XCL. Each chemokine has its corresponding chemokine receptor [5,6].

### 5.2. Immune Cell Recirculation

Immune cells are traveling most of their time. Starting from bone marrow (BM), they reach the blood, from which they transmigrate into tissues. Distinct types of immune cells target specific compartments in a process called homing. Also, in tissues, the immune cells remain mobile. They migrate from tissues via afferent lymphatic vessels into the draining lymph node. From there they can leave via efferent lymphatics. After visits to several lymph nodes, the cells enter the thoracic duct, which drives them via the vena cava back into the blood circulation [5]. Lymphocyte recirculation allows naïve lymphocytes (B and T cells) to continuously patrol the body’s sites of antigen entry, even in the absence of inflammation.

Different lymphocyte subsets show differences in recirculation patterns and lymphocyte homing. These correlate with the expression of varying adhesion molecules on the surfaces of the lymphocytes and on the endothelial venules of the various tissue sites. Adhesion molecules expressed selectively on high endothelial venules (HEVs) of lymph nodes are called addressins, and the complementary adhesion molecules on the lymphocytes are called homing receptors [5,6].

Here are a few examples of innate and adaptive immunity cells. Monocytes from the BM, circulating in blood, emigrate through capillary walls into tissues where they may differentiate into macrophages or myeloid dendritic cells (DCs). Granulocytes (more than 90%) remain in the BM as a reserve. In case of tissue inflammation, they are recruited to the target site where they execute their effector function before dying. DCs, distributed throughout the body, constantly check their local environment for the presence of self antigens (SAs) or non-self antigens (NSAs) [76]. Naïve T cells that survived the selection processes in the thymus are released into the blood circulation. They travel through the blood and the lymphatic system. As soon as they detect in lymphoid organs DCs presenting neoantigens and danger signals, they react to these professional antigen-presenting cells (APCs), become activated, and differentiate into clones of effector T cells and memory T cells (MTCs). Such activated T cells have the capacity to emigrate from blood via capillaries into peripheral tissues to search for their target. Interestingly, MTCs that detected their neoantigen in lymph nodes draining the skin home again to the skin, while MTCs that detected their neoantigen in a mesenteric lymph node migrate and home to the respective mucosa [5,6].

### 5.3. Mechanisms of Lymphocyte Homing

Molecular details of T cell homing from blood to tissues include rolling (via selective lectins (selectins) and their ligands), activation (chemokines and their receptors), adhesion (integrins and their ligands), and diapedesis through the endothelium. Each immune cell “knows” where to leave the blood system. Example 1: IgA+ B cells target mucosa via the following three codes: (i) fitting selectin-selectin ligand interactions; (ii) fitting chemokine-chemokine receptor interactions; and (iii) fitting integrin-integrin ligand interactions. Example 2: Naïve T cells expressing L-selectin (CD62L) at high density at their surface interact with the L-selectin ligand peripheral node addressin (PNAD) of HEVs of lymph nodes. The circulating T cells slow down and roll along the HEV epithelium. Since T cells express CCR7 chemokine receptors, they receive activation signals through the chemokine CCL21, which is expressed constitutively by HEVs, and through CCL19, which comes from lymph node cells. Such interactions lead to a conformation change in the integrin lymphocyte function associated antigen 1 (LFA-1) on T cells, which enables a high affinity interaction with the ligands intercellular adhesion molecule (ICAM)-1 and ICAM-2 on HEVs. This causes arrestation followed by diapedesis of the T cells into the T cell area of the draining lymph node [5,6].

What is less known is that naïve mature T cells home not only to HEVs of lymph nodes but also to flat endothelium of sinusoidal blood vessels in BM [77,78]. Circulatory antigen-presenting DCs that might express SAs or blood-borne neoantigens as NSAs also home to BM. Ligands known to be mandatory for homing of T cells and DCs to BM are vascular cell adhesion molecule 1 (VCAM-1), mucosal addressin cell adhesion molecule 1 (MadCAM-1), and ICAM-1 for T cells and ICAM-2 for DCs. These are constitutively expressed in BM stroma [76,77]. After diapedesis of BM flat endothelium, the T cells and DCs enter the spongyform BM parenchyma with their many specific niches [76,77,78] in the search for cognate interactions. The 3D microanatomical organization of BM has been described [79].

Hematopoietic and osteogenic stem and progenitor cells, memory B and plasma cells, and memory T cells home to specific niches in BM [76,77].

The expression of all mentioned adhesion molecule families is regulated by immune and endothelial cells. In case of an inflammation, inflammatory cytokines induce adhesion molecules and chemokines on endothelial cells so that more immune cells from the blood can be recruited [5]. Intercellular adhesion between immune and non-immune cells is facilitated by 24 known integrins consisting of one (of 18) alpha chain and one (of 8) beta chain [5,6].

### 5.4. Importance of the Central Immune System Bone Marrow

The bone includes a series of blood vessels organized in a specific order to provide nutrients, regulatory factors, and oxygen to the cortex and medulla. Blood flow also removes metabolic waste products such as carbon dioxide and acid [80]. BM sinusoids form a capillary network with venous sinusoids, and the latter converge to a large sinus in the BM center [80]. Blood exits the medulla via multiple small veins that penetrate the cortex [81].

BM tissue inside the different long, short, and flat bones constitutes one of the largest organs in humans, accounting for 4–5% of the total body weight (TBW). In comparison, the entire network of secondary lymphoid organs makes up only 1–1.5% of TBW. BM is the most prominent source of de novo cellular generation, reaching rates of 4–5 × 10^11^ cells per day in an adult human. At any given moment, 90% of the total cells in the organism originate from or reside in the BM, with anuclear erythrocytes and platelets accounting for the majority [79]. Like the brain, BM has a rich and sophisticated microanatomical organization [79].

### 5.5. Summary

In summary, spatial orientation in the immune system functions similar to a postal code. T cells home to the site where they once recognized an antigen. Gradients of chemokines guide cells with respective receptors to the site where the chemokine is released. Adhesion molecules and respective ligands allow the blood circulatory cells to dock to the endothelium and to transmigrate into tissues. A coordinated action of chemokines and adhesion molecules guides hematopoietic stem cells and B and T memory cells to distinct niches in the BM for long-term survival.

## 6. Immune System: Learning

### 6.1. Initial Training of Lymphocytes

Lymphocytes express highly diverse antigen receptors that are capable of recognizing a wide variety of foreign substances. Positive and negative selection processes [5,6] of T lymphocytes in the thymus and of B lymphocytes in the bone marrow represent a kind of initial training for lymphocytes not to attack self antigens and learn to react only to non-self (foreign) antigens.

### 6.2. Fighting Infections

In 1885, L Pasteur injected the first inactivated rabies vaccine into a 9-year-old boy. Twenty years later, in 1905, R Koch received the Nobel Prize in Medicine for his discoveries about tuberculosis. As a physician in a praxis in Posen, he saw every day the inefficiency of the existing art of medical healing against infectious diseases, such as anthrax, cholera, tuberculosis, malaria, sleeping sickness, and pest. Both medical research pioneers can be considered the founders of active specific immunization (ASI) or vaccination.

Active immunization can occur naturally (unintended) or artificially (deliberate). Natural passive immunization occurs by transfer of antibody from mother to infant in placental circulation or colostrum. Passive antibody therapy can be serum therapy, administration of immune human globulin, or, nowadays, of monoclonal antibodies (mAb). Also, bispecific antibodies (bsAb) and trispecific immunocytokines (tsIC) have been developed, e.g., to target the immune system against cancer [82]. bsAbs and tsICs are recombinant proteins with two or three specific binding sites from mAbs or cytokines, capable of stimulating effective immune responses.

### 6.3. Immune Education by Vaccination

Protection against infectious diseases by the use of vaccines represents perhaps the greatest accomplishment of biomedical science. One disease, smallpox, has been totally eliminated by the use of vaccination, and the incidence of other diseases has been reduced significantly [5,6].

Humans become educated in skills such as speaking a language, reading, or playing a musical instrument. In a similar way, the immune system can become educated deliberately, for instance by vaccination.

Successful vaccines can be (i) killed whole organisms (e.g., against typhoid), (ii) attenuated bacteria (e.g., Bacillus Calmette-Guérin (BCG) against *M. tuberculosis*), (iii) toxoids (e.g., against diphtheria or tetanus), (iv) surface molecules (e.g., against influenza), (v) inactivated virus (e.g., Salk vaccine against polio), (vi) attenuated virus (e.g., Sabin oral polio vaccine and vaccines against childhood diseases such as measles, mumps, or rubella), (vii) recombinant viral proteins (e.g., against hepatitis B or human papillomavirus).

Additional vaccinations can be applied to selected human populations. For instance, military personnel (meningococcus), persons in contact with rodents (plague), veterinarians, animal handlers (rabies), travelers to high-risk areas (typhoid, yellow fever).

Immunity can be induced against bacteria (e.g., gram-positive, Gram-negative, mycobacteria, spirochetes), viruses (e.g., hepatitis virus, coronavirus), parasites (e.g., protozoa, helminths), and fungi (candida albicans) [5,6].

Immunity can also be induced against internal antigens derived from blood (blood-borne antigens) or cerebrospinal fluid (CSF). An important example is cancer. This, however, is more difficult and sophisticated than the elicitation of immune reactions against foreign agents. Neoantigens are tumor-specific antigens (TAs) derived from genetic, transcriptomic, and proteomic alterations unique to cancer cells [83]. Anti-cancer vaccines can be based on polypeptides or proteins expressing defined neoantigens or on plasmid DNA, recombinant viral vectors, and prime-boost immunizations. Dendritic cell-based anti-cancer vaccines are another modality that is directed towards activating CD4 T helper type 1 and CD8 T cytotoxic T lymphocytes (CTLs) [5,6].

Learning in school (brain activity) can be influenced positively by attention, emotion, motivation, and repetition. Repetition is also of importance for learning in the immune system. Repeated vaccinations can lead to consolidation of long-term memory [5,84].

Long-term protective anti-tumor immunity can be induced by repeated cycles of vaccination with cancer cell or DC vaccines modified by virus infection [84]. This is particularly effective if the virus is an oncolytic virus such as the avian Newcastle disease virus (NDV) because this virus has the capacity of breaking therapy resistance [85]. The scientific rationale and clinical experience of anti-tumor vaccination with NDV-modified DC vaccines from a single institution in Germany have recently been published. It represents an individualized multimodal immunotherapy coined IMI [86]. A positive example is the results obtained in the treatment of glioblastoma multiforme (GBM) [86].

### 6.4. Gut Microbiota

Our gut contains 1 to 1000 billion bacteria and other microbes, which have important functions. This microflora contains a variety of different species that colonize different parts of the gut, starting from the esophagus (e.g., streptococcus), via the stomach (e.g., helicobacter), duodenum (e.g., lactobacillus), jejunum (e.g., enterococcus), ileum (e.g., clostridium), and colon (e.g., klebsiella) towards the anus (e.g., staphylococcus). During a natural birth, a child becomes exposed to the mother’s gut microbiota, which stimulates its immune system.

The child’s immune system needs time to develop competence. During this developmental period, it has to learn to differentiate, among others, between bacterial “friends and foes”. This has a lifetime perspective.

Gut microbiota (GM) live in symbiosis with the human body and help digest the food taken in. This coexistence between bacteria and the vertebrate’s gut is of advantage for both sides. An energy source and protection for the bacteria and help with digestion for the host. The microflora is capable of adapting to the diet habit of the host. An additional function of gut microbiota is to protect against pathogenic bacteria. Also, gut microbiota–immune–brain interactions maintain brain homeostasis and influence brain and behavior [87]. The microbiota–gut–brain axis facilitates communication via metabolites such as short chain fatty acids, neurotransmitters such as dopamine and GABA, and hormones like serotonin and melatonin [88]. Gut commensal bacteria-specific CD4 T cells that are dysregulated in the inflamed gut can become licensed to infiltrate into the CNS, whereupon they produce high levels of GM-CSF, IFN-gamma, and IL-17A, triggering neurological damage [89].

## 7. Immune System: Memory Function

This chapter is subdivided into eleven sections: Innate and adaptive immune memory (Section 7.1 and Section 7.2), MTCs, their diversification, dynamics and longevity (Section 7.3, Section 7.4 and Section 7.9), the importance of BM (Section 7.5 and Section 7.7), and immunotherapy studies with MTCs (Section 7.6 and Section 7.8), molecular mechanisms in immune memory (Section 7.10) and summary (Section 7.11).

### 7.1. Innate Immune Memory

In the science of immunology, memory was traditionally considered an exclusive hallmark of adaptive immunity. This dogma was challenged by recent reports that myeloid cells can retain memory of earlier challenges, enabling them to respond strongly to a secondary stimulus [90]. This process, designated “trained immunity”, is initiated by modulation of precursors of common myeloid progenitor (CMP) cells in the BM. Changes in cellular metabolism are suggested to be a common denominator of innate immune memory from lower animals to mammals [90].

### 7.2. Adaptive Immune Memory

Antigen-specific B cell responses produce memory cells of two kinds: specialized B memory cells and long-lived plasma cells. Clonal selection and expansion after antigen contact lead to a large increase in the frequency of antigen-specific B cells compared to the naïve B cell repertoire. Differentiation to memory B cells is associated with a change in the class of the BCR, namely from IgM to IgG. B cell memory functions systemically since the secreted antibodies distribute throughout the organism. During the course of a B cell response, the affinity of the antibody to the antigen increases due to a somatic affinity maturation mechanism in secondary lymphatic organs. An example of the long-term efficacy of immune cell memory is the measles virus epidemic from 1781 on the islands of Faroe in the North Atlantic. During a second epidemic in 1848, none of the people beyond the age of 64 were affected [5,6].

Following cognate interaction between a naïve T cell and an APC expressing an antigen fitting to the TCR of the naïve T cell, a synaptic supramolecular membrane activation complex (SMAC) is formed, followed by a massive T cell clonal expansion. At the peak of the effector phase after vaccination against smallpox with vaccinia virus vaccine, 2–4% of all CD4 T cells and 5–15% of all CD8 T cells in peripheral blood were vaccine specific. This represents a 200–10 000 fold expansion of antigen-specific T cells. Three months later, 5% of the antigen-specific CD4 and 3% of the antigen-specific CD8 effector T cells had differentiated into MTCs [5,6].

Termination of effector T cell responses is executed through activation-induced cell death (AICD), a form of programmed cell death (PCD), cytokine (e.g., IL-2) “withdrawal”, and T cell clonal exhaustion [6].

### 7.3. Memory T Cells

Memory differentiation of CD4 and CD8 T-cell populations has been extensively studied [5,6,84,91] and many key molecular patterns and transcription factors (e.g., Thpok [92], Runx [93], c-Myb [94]) have been identified. The first few days of T-cell expansion may be particularly conducive to transcriptional and epigenetic plasticity [91]. During these early times, T-cell families are still small, and the behavior of a few daughter cells may shift the developmental balance of the complete family [91]. This is reminiscent of the early phase of WM in the brain. Upon antigenic challenge, MTCs respond faster than naïve T cells, and they secrete a broad spectrum of cytokines (IFN-gamma, IL-4, IL-5, IL-10, IL-17, TGFß) in contrast to the antigen-specific naïve T cells, which only secrete IL-2 [5,6]. Naïve T cells express on their membrane the long form of CD45 (CD45RA), while MTCs express a smaller splice variant (CD45R0). These are suitable markers for distinction. MTCs express a different spectrum of adhesion molecules and chemokine receptors compared to naïve T cells. This allows them to transmigrate into tissues where they execute their specific surveillance function [5,6].

There exist two major classes of MTCs: effector MTC (eMTC) and central MTC (cMTC). Differences in homing receptor expression between these populations lead to differences in their distribution throughout the body. cMTCs express high levels of the lymph node homing receptors CD62L and CCR7 and migrate through secondary lymphoid tissues. In contrast, eMTCs express only low levels of CD62L and CCR7 so that they circulate mainly through non-lymphoid tissues. Such tissues include the lung, intestines, reproductive tract, liver, and fat [6]. Bone marrow tissue-resident MTCs (bmMTCs) are a recently described new subset of MTCs [95,96,97,98]. They were detected, among others, in murine and human BM. Being dependent on IL-15, these T cells reside long term in special niches of BM that provide IL-15 [99].

Another type of MTC is the interesting subset of stem cell-like MTCs (sMTC), characterized by cell surface expression of CD69 and CD127 [97]. These naïve-like MTCs show preferential homing to BM [98]. Interestingly, sMTCs, upon adoptive transfer, are strongly resistant to tolerance induction. One study used IL-7 and IL-15 to instruct the generation of human sMTCs from naïve precursors in vitro [99]. Another recent study provides a new technique for gene repair and expansion of autologous sMTCs for adoptive transfer to patients with T cell immunodeficiencies [100].

Massive proliferation and differentiation into diverse long- and short-lived T cell subsets are required to provide acute protection against primary infection and lasting immunity against reinfection with the same pathogen. An interesting single-cell approach to MTC differentiation has recently been summarized [91]: A single naïve precursor T cell, transferred to an immunocompetent mouse recipient, generates a diverse primary immune response to an acute bacterial infection in vivo. The multipotency of cMTCs is demonstrated by the ability of a single primary cMTC to generate a diverse secondary immune response. The capacity for self-renewal of cMTCs was tested by the ability of a single secondary cMTC to generate a diverse tertiary immune response. This second adoptive transfer experiment established the capacity for self-renewal of the primary cMTC. The study demonstrated robustness of T-cell immune responses by averaging distinct single-cell behaviors [91]. Differential expansion of cMTC precursor and effector subsets was reported to be regulated by division speed [101]. The exact drivers of diversification in vivo were suggested to rest on asymmetric cell division, quorum sensing, microanatomical variation in instructive signals, or stochastic transcriptional noise [91].

### 7.4. Dynamics and Longevity of T Cell Memory

Dynamics and longevity of T cell memory were studied in a mouse model using the bacterial lacZ gene product ß-galactosidase (Gal) as a surrogate tumor-associated antigen (TA) and lacZ transfected ESb (ESblacZ) tumor cells for tumor challenge experiments in T cell deficient nude (nu/nu) mice [102]. Polyclonal Gal-specific eMTCs from immunocompetent mice were transferred together with live ESblacZ tumor cells intraperitoneally (i.p.) into nude mice. While naïve nude mice died within 10 days following tumor cell injection (1 × 10^5^ cells), eMTC-transferred nude mice were able to reject a tumor dose of up to 5 × 10^7^ and survived longer than eight months [102]. The eMTC transfer resulted in long-term persistence of Gal-specific T cells in BM and spleen. The processes of tumor cell restimulation (eMTC generation within 3 days), eMTC transfer followed by a resting period (about 2 months), repeated up to the sixth antigen contact in quaternary nude mouse hosts, demonstrated longevity of T cell memory. Interestingly, repeated tumor cell challenge was necessary for long-term maintenance of T cell memory. Waves of activation in the peritoneal cavity (eMTCs) were followed by changing to cMTCs and resting phases after homing to BM niches, demonstrating the dynamics of the memory response [102].

The study suggested, among others, that persistence of memory lymphocytes requires a periodic low level stimulation by tiny amounts of residual antigen. Such stimulation might help to induce the expression of base levels of anti-apoptotic molecules that would permit the MTC to survive [6].

### 7.5. Bone Marrow as a Priming Site for T-Cell Responses to Blood-Borne Antigen

A first hint to the potential importance of the BM in immunosurveillance came from experimental studies in mice [103]. These demonstrated active control of proliferating tumor cells by CD8 MTCs, which led to tumor dormancy in BM. Further studies revealed that BM parenchyma contains DCs capable of presenting antigens to naïve CD4 [104] and CD8 [105] T cells. Cognate cellular interactions resulted in T-APC immunological synapse formation, T-APC cluster formation, and primary BM T cell responses within 7–9 days [105].

### 7.6. Clinical Studies with Cancer-Reactive Memory T Cells from Bone Marrow

The induction and maintenance of T cell memory by the BM microenvironment and clinical studies with reactivated BM-derived cancer-reactive MTCs have been reviewed [106]. It is worth mentioning a pilot clinical adoptive immunotherapy (ADI) study based on bmMTC transfer in advanced metastasized breast cancer [107]. The results of a follow-up analysis showed that patients with an immunological response in the peripheral blood (as a result of massive in vivo MTC expansion) had a significantly longer median survival than nonresponders [108].

### 7.7. Bone Marrow Functioning as a Refuge for Memory T Cells in Situations of Energy Crisis

Major chronic diseases and hunger crises are characterized by mitochondrial dysregulation of the cellular energy supply and metabolism. In experimental studies in mice, short-term fasting (dietary restriction, DR), creating hunger, had effects on MTCs [109]: MTCs collapsed in secondary lymphatic organs and accumulated within the BM. Thus, BM was reported to function as a refuge for immune memory in situations of energy crisis [109]. This mechanism apparently developed during evolution with an advantage for survival; otherwise, it would not have been selected. Details about the effects of DR on the immune system (monocytes, B and T cells) have been summarized [76].

### 7.8. Adoptive Cancer Immunotherapy with MTCs from BM in Mice

An extreme deviation from homeostasis is seen in late-stage cancer, particularly when associated with cachexia (weakness and wasting (weight loss) of the body due to severe chronic illness). There exists no therapeutic option. ADI in this situation was evaluated in a graft-versus leukemia (GvL) mouse model [110]. The first step consisted of priming donor strain mice (B10.D2; MHC class d) against the cancer (ESb leukemia) of the recipient strain (DBA/2, MHC class d) and harvesting donor BM-derived MTCs. A single adoptive transfer of cancer-reactive MTCs from the BM of such donor mice into 5 Gy irradiated late-stage tumor bearing recipients led to effective immune rejection of advanced disease with cachexia [110]. An approximately 25,000 fold excess of metastatic tumor cells was rejected, as revealed quantitatively by FACScan analysis of lacZ gene transfected tumor cells [110]. An analysis of mechanisms of this ADI strategy revealed that the strong GvL effect was achieved, among others (involvement of multiple types of cellular interactions, granzyme B, perforin, TRAIL, NO), by breaking immunological tolerance to a tumor-associated endogenous viral superantigen (i.e., vSAG-7) [111]. Tumor-reactive MTCs from the BM of the donor mice exerted GvL without graft versus host (GvH) reactivity [112]. This most effective adoptive MTC transfer is a good example (i) of the potential strength of an MTC-mediated anti-tumor immune response, (ii) of principal reversibility of late stage cancer, and (iii) of return from cachexia to homeostasis.

Adoptive cancer immunotherapy can also be performed with induced pluripotent stem cell derived engineered T cells, natural killer cells, macrophages, and DCs [113].

### 7.9. Diversification of MTCs

The diversity of MTCs generated from single T cells in vivo [91] as well as the dynamics and longevity of Gal-reactive T cell memory [102] suggest clonal cooperation. Categories characterizing an immune object such as Gal may be antigenic strength, context, temporal and localization signals. An integrated perception of an immune object may be first perceived by distinct networks of immunocompetent cells [114]. The output signals generated during this primary perception step might then be integrated by T helper cells at the single cell level. Then, in a multitude of lymphoid organ niches, further processing eventually generates a multitude of T-cell and B-cell clones that perform an integrated perception of immune objects. This point of view has been inspired by the visual system, in which distinct features of a visual object (shape, color, motion) are perceived separately by distinct neuronal populations during a primary perception task. The output signals generated during this first step then instruct an integrated perception task performed by other neuronal networks [114].

Another theory suggests modifying the clonal selection theory of the Nobel laureate of 1960, the immunologist F.M. Burnet, with a probabilistic cell. The theory aims to explain immunity in terms of cell population dynamics [115]. Shaping heterogeneity of naïve CD8 T cell pools by various intrinsic and extrinsic factors has been suggested to regulate CD8 T cell activation, proliferation, and differentiation into effector and memory cells. These include TCR signaling via SAs and homeostatic cytokines such as IL-7 and IL-15 [116].

### 7.10. Molecular Mechanisms in Immune Memory

Coming to molecular mechanisms, emerging principles focus on transcriptional memory [117] and on microRNA regulation of CD8 T cell responses [117,118]. Transcriptional memory is characterized by a primed cellular state, induced by an external stimulus that results in altered expression of primed genes upon re-exposure to the inducing signal. The primed state is heritably maintained across somatic cell divisions even after the initial stimulus and target gene transcription cease. This phenomenon is widely observed across various organisms and appears to enable cells to retain memory of external signals, thereby adapting to environmental changes [117]. Key molecular signatures include the following: the poising (outbalancing) of RNA polymerase II machinery, maintenance of histone marks, and alterations in nuclear positioning and long-range chromatin interactions [117]. MicroRNAs (miRNAs) are a class of short non-coding RNAs that play critical roles in the regulation of a broad range of biological processes. Like transcription factors, miRNAs exert their effect by modulating the expression of networks of genes that operate in common or convergent pathways. How this might function in CD8 T cells as critical agents of the adaptive immunity system, has been reviewed [118].

Non-coding RNAs (ncRNAs), including miRNAs, long non-coding RNA, and other ncRNAs, are functionally important for learning, memory, and adaptive immunity [119]. They also regulate experience-dependent neural plasticity [120]. ncRNAs are highly plastic RNA molecules that can sequester cellular proteins and other RNAs, serve as transporters of cellular cargo, and provide spatiotemporal feedback to the genome. A fundamental neurochemistry review states that mounting evidence indicates that ncRNAs are central to biology, and are critical for neuronal development, metabolism, and intra- as well as intercellular communication in the brain and in the immune system [120]. For evolutionary aspects of the RNA world, see Section 8.9.

### 7.11. Summary

In summary, memory functions in the immune system are exerted through innate and adaptive immunity cells. Niches in BM provide sites for long-term survival of memory lymphocytes. Billions of cells per day circulate between BM and blood. Important examples of immune memory are the protective responses against measles virus and smallpox virus infection. Massive proliferation and differentiation into diverse long- and short-lived T cell subsets are characteristic of immune system responses in comparison to neuronal responses. The diversity, flexibility, cell population dynamics, and longevity of T cell memory suggest clonal cooperation and an integrated perception of immune objects. With regard to molecular mechanisms, emerging principles focus on transcriptional memory and on non-coding RNA regulation of CD8 T cell responses. Mounting evidence indicates that ncRNAs are critical not only for central biological processes (development, metabolism, intra- and intercellular communication) in the immune system but also for the brain. The first few days of T cell response expansion may be particularly conducive to transcriptional and epigenetic plasticity, in similarity to the early phase of WM in the brain.

## 8. Discussion

### 8.1. Brain: Spatial Orientation

The hippocampal formation (see Figure 1) is a phylogenetically ancient and well-preserved structure. It is known to provide a detailed representation of an animal’s location and heading. Neuroscientific research over the last four decades has identified distinct types of neuronal cells in the hippocampus that provide spatial information: place cells, head direction cells, boundary vector cells, and grid cells. The joint activity of grid cells from an individual module was recently described to reside on a toroidal manifold, which is a 3D space. Brain oscillations seem to play an important role in grid cells toroidal topology.

The trisynaptic circuit is a relay of synaptic transmission that is made up of the following three major cell groups: granule cells in the DG, pyramidal neurons in CA3, and pyramidal neurons in CA1. This circuit is a key part of spatial orientation, learning, and memory.

Plasticity and flexibility are important during evolution to adapt to changes in the environment. Circuits in the hippocampus are indeed remarkably plastic. Such plasticity can be mediated by changes in synaptic strength, as revealed by long-term potentiation and long-term depression. Changes can also occur in neuronal firing rates, as exemplified within the theta rhythm.

Spatial orientation occurs not only via optical signals. It may arise as well from acoustical, olfactory, magnetic, or other signals. Haptic and chemical communication occurs for orientation in the dance language of bees and in the pheromonic world of social insects. Examples of acoustical orientation are freely flying bats; of olfactory orientation, snakes and moles; and of magnetic orientation, migratory songbirds and pigeons.

Jellyfish is an example of sea animals without brains and spatial orientation. Deep sea animals can use a combination of senses to orient themselves, including sight, sound, and possibly magnetism. Vertical migration helps their orientation in response to light, temperature, and salinity. An astounding sea animal is the octopus, a mollusk belonging to the class of Cephalopoda. Its brain structure is distributed from the head to the eight arms. With these multiple mini brains, such animals are capable of fast orientation, learning, and memorizing, all signs of intelligence.

### 8.2. Brain: Programmed Learning

The term “imprinting”, coined by K Lorenz, refers to cases of “programmed learning” during a critical time period. In geese this period of particular sensitivity to experience is limited to a few hours after birth.

During embryonic development, human brain evolution is “imprinted” by a neurogenic period in the sixth month of pregnancy. This period programs neuronal stem cell (NSC) division rate and duration of the neurogenic period [121].

In the first part of this review [2], a unifying homeostasis hypothesis has been presented concerning neuroimmune crosstalk between the central nervous system (CNS) and the central immune system (CIS) [76]. The hypothesis states that this homeostasis depends on the regulation of stem cells that reside in special niches of BM and which are involved in the generation of these tissues. The relevant stem cells are HSCs for the immune system, NSCs for the CNS, and MSCs for the bones [76].

NSCs play a decisive role in evolution of the large human neocortex.

### 8.3. Evolution of the Human Neocortex

The neocortex, evolutionarily the youngest part of the brain, is the seat of higher cognitive abilities. The human neocortex exhibits an increase in size and in the number of neurons compared to non-human primates [121]. This increase is proposed to reflect a greater proliferative capacity of the cortical stem and progenitor cells (cNPCs) in humans [121]. Human cerebral organoids have been shown to recapitulate gene expression programs of fetal human neocortex development [122]. Over the past ten years, genes have been identified that specifically evolved in the human lineage and are preferentially expressed in cNPCs [122]. For example, the human-specific gene ARHGAP11B has been implicated in human neocortex expansion [122]. A mutation in this gene, which occurred about 2 million years ago, has been suggested to have led to the enlargement of the human neocortex. This mutation has the effect that the gene product moves into mitochondria, where it stimulates division of cNPCs. More neurons and more wiring means more primate-specific cortical complexification and encephalization [123].

If this hypothesis were true, the scientific implications were enormous: 7 million years ago, the size of the brain of the common precursor of humans and chimpanzees was 300–400 cm^3^. Five million years ago, occurred a partial duplication of gene ARHGAP11A, which created gene ARHGAP11B. The brain size of Australopithecus africensis, 4 million years ago, was about 450 cm^3^. The above mentioned point mutation in ARHGAP11B occurred in homo habilis, who had a brain size of about 600 cm^3^. It was a period of environmental change when homo habilis left the forests and walked upright in the savanna. Homo erectus, who lived about 1 million years ago and who entered Eurasian territory, had a brain size of about 1000 cm^3^. Homo sapiens today has a brain size of about 1300 cm^3^.

It is likely that more than one gene was associated with this brain development. PDYN, an endorphin that encodes the neurotransmitter prodynorphin, with its cis-regulatory variation, could be another gene example [124]. Upon binding to opiate receptors, this endorphin mediates our experience of pain, social attachments, learning, memory, and addiction.

A third example is the transcription factor FOXP2. It has been suggested as a molecular window into speech and language [125]. Reduced FoxP2 dosage yielded abnormal synaptic plasticity, impaired motor-skill learning in mice, and disrupted vocal learning in songbirds [125].

Thus, rare mutations in distinct genes (ARHGAP11B, PDYN, FOXP2) might have had an impact on the evolutionary path from homo habilis to homo sapiens. Since real evidence is lacking so far, this information has been incorporated into the discussion.

I propose in this discussion additional mechanisms: genetic elements can become multiplied, and their position in the genome can be changed by means of transposons and retrotransposons. Such mechanisms could have had impacts during evolution, including the changes in the size and complexity of the human neocortex.

### 8.4. Development of Human Culture: Writing, Arts, Worshiping, and Future Prediction

The capability of writing developed in homo sapiens in the 4th to 3rd millennium before Christ, independently three times, in Egypt, Mesopotamia, and China. This enabled the transfer of memory from the brain into texts and the accumulation and conservation of them for future generations. Writing capability distinguishes prehistoric from historic times. In old historic times, prophets of the Bible from Israel and Judaea tried to convey God-mediated revelations about the future to the royal court and to advise it about military, economic, or political questions. During the Bronze Age, Homo sapiens developed high cognitive abilities, leading to typical human culture, such as writing, performing arts, worshiping, and trying to make associations between past, present, and future events.

The capability to outsource knowledge from human memory into written texts, to accumulate, edit, and keep them for future generations opened up new possibilities of human development, which expanded with a hitherto unknown speed. The Bible is a prototype example of a text written by homo sapiens, translated into more than 560 languages, and printed in approximately 5 billion copies. It is the most well-known and widespread book of world literature.

### 8.5. Brain: Memory Function

All the cognitive abilities described in Section 8.4. are human-specific and distinguish homo sapiens from the animal kingdom. They were facilitated by an enlarged neocortex. This allowed memory functions of immense complexity. This review could only describe a few basic principles of memory function. In the brain, engram neurons are involved in encoding, consolidation, and retrieval. During memory recall, many of the same cells are reactivated. The dorsolateral prefrontal cortex is implicated in working memory in addition to emotional and cognitive processing.

The CA1 output region of the hippocampus plays an essential role in the retrieval of top-down formation of episodic memory. In the post-encoding period, bottom-up transfer to lobes of the cortex is involved in the formation of semantic memory. Information processing includes exchange between the prefrontal cortex and temporal cortex (top down) and vice versa (bottom up). During sleep, TMR enhances bottom up memory consolidation by presenting reminder cues.

Acquired memory is achieved by synaptic connectivity in groups of pyramidal neurons in the neocortex. Spatial orientation and spatial memory are embedded in electromagnetic field oscillations and resonance effects produced by the brain.

Figure 1 integrates sites of spatial orientation, learning, and memory function deep in the temporal lobe. Other lobes are involved in information processing. The figure shows lobes of sensory input from vision, hearing, and smelling. Other sensory inputs come from taste, thermosensation, mechanosensation, electroreception, the magnetic sense, and pain and nociception. Each sensory input has its own sites for information processing [4].

### 8.6. Immune System: Spatial Orientation

Unlike the brain, with its mostly immobile network of neurons, the immune system is based on a network of mostly mobile cells. For instance, the task for a mature naïve T cell to find an APC with an antigen fitting to its receptor is like the task of finding a needle in a haystack. To solve this problem, nature has developed lymphoid organs with distinct areas for T or B cells to home to. Secondly, it has invented guides, namely chemokines and respective ligands and receptors. If a target site within a lymph node releases a certain chemokine ligand, a T cell expressing the corresponding chemokine receptor becomes triggered to follow the track according to the concentration gradient of the ligand. APCs are attracted nearby so that the chances for cognate interactions are much higher.

### 8.7. Immune System: Learning

An important task of the immune system is to distinguish between self and foreign antigens and between signals mediating tolerance or danger. Both, innate and adaptive immunity systems contribute to decision making. A specialty of the adaptive immunity system is the process of somatic gene rearrangement. This allows each B or T lymphocyte to express a unique antigen-specific receptor at its cell surface. These receptors are pre-selected in lymphatic organs (thymus, BM) against self-reactivity. The repertoire of B and T cell receptors is enormous. Any foreign chemical structure, be it natural or artificial, can become recognized to elicit an immune response.

The cognition of a neoantigen (i.e., a pMHC complex) by a T cell expressing a fitting receptor requires neoantigen presentation by APCs such as DCs. Cognate T-APC interactions initiate the formation of an immunological synapse in which chemical signals are being exchanged. This intercellular communication is similar in complexity to that of brain synapses. Similarities include endosome recycling, exocytosis mechanisms, and secretion of informative structures (neurotransmitters and synaptic ectosomes, respectively) into the synaptic cleft [2]. The theoretical number of different T-APC synapses is in the range of or beyond that of the synapses in the brain. In addition, there are the B cells recognizing soluble antigens directly with their enormous repertoire of diverse receptors.

### 8.8. Immune System: Memory Function

An immune response is characterized by a clonal expansion phase, an effector phase, a retraction phase, and a memory phase. At the end, only a few percent of the antigen-specific cells survive, but these are of high relevance for protection against any further exposure to the same antigen (e.g., an infectious agent). Memory B and memory T cells have distinct survival niches in the BM. There also exist interesting stem-like memory T cells in the BM. Other memory subsets are tissue-resident or travel through the blood circulation. Upon reactivation, central memory T cells differentiate quickly into effector memory T cells to execute their protective function. A single-cell approach to memory T cell differentiation revealed in vivo a capacity for diversification. Other studies demonstrated a high degree of dynamics and longevity of T cell memory. The findings suggest clonal cooperation and flexibility.

An immune memory recall response differs from a neuronal memory recall response in that it goes through a massive cellular expansion phase, which can be 200–10,000 fold. Otherwise there are similarities in memory function. A priming phase in the immune response corresponds to an engram phase in neurons. In both systems, consolidation takes place through synaptic and network communication. Memory responses in both systems can be repeated many times. Both systems are characterized by openness. They have the capability to react and to respond to the unforeseen event or to a new, hitherto unknown infectious agent.

Figure 2 summarizes key points of the immune system and of communication sites (neuroimmune interfaces) between the brain and the immune system. Studies of recent years have described immune dynamics in the CNS and its barriers during homeostasis and disease [126]. A recent editorial suggests a neuroimmune web of pain, mood, and memory [127]. It thus enlarges our knowledge about neuroimmune interactions. Neuroimmune stress circuitry can lead to mitochondrial dysfunction, depression, and neurodegeneration like Alzheimer’s. Natural compounds such as curcumin and flavonoids are reported as promising multitarget tools for mood modulation. Anti-CGRP/R monoclonals (mAbs) targeting sensory peptides apparently soften migraine, depression, and anxiety [127]. Dysfunction of mitochondria has been implicated in the pathogenesis of various brain diseases, including neurodegenerative disorders, stroke, and psychiatric illnesses. A comprehensive review of mitochondria and brain disease describes pathological mechanisms and therapeutic opportunities [128]. Wherever appropriate, the two parts of this review mention dysfunctions and neuroimmunomodulatory procedures of treatment.

Figure 2 also points towards the importance of BM. Cancer-reactive MTCs, especially those from BM, are a promising source for adoptive T cell immunotherapy. This was demonstrated by the author and colleagues in animal models (including late-stage disease), in xenograft studies with cancer patient derived cells, and in a phase I clinical study in advanced breast cancer patients.

### 8.9. Infections: Evolutionary Aspects

One study investigated whether present-day variation in a key component of the immune system represents evolutionary adaptation to ecological factors such as parasites, injuries, and predators [129]. It was concluded that the risk of disease infection from the environment and the risk of injury have played a key role in immune system evolution among anthropoid primates [129].

Bacteria, protozoa, and fungi lurk everywhere. There are more “bugs” grazing on or in a single human body than there are people in the entire world [130]. Viruses are much more demanding than the above “bugs” because to survive and reproduce, they must penetrate a living cell. D Crawford has described many aspects of the natural history of viruses as deadly parasites, explaining how they differ from other microorganisms [130].

It is obvious that during evolution the fight of the immune system against viruses (e.g., influenza, HIV, Ebola) has been a major challenge. Education of the immune system by targeted vaccination with a new type of vaccine, mRNA vaccine, has helped mankind against the recent SARS-CoV2 pandemic. Viruses, as old as 3.5 billion years on Earth, could not only have been deadly parasites. They also could have been beneficial for the evolution of vertebrates. This is at least the personal view of the virologist K Moelling [131]. She studied reverse transcription in retroviruses such as HIV. In this context it is interesting that the remains of human endogenous retroviruses (HERVs) make up as much as 8% of our DNA [132]. It has been suggested recently that endogenous retroviruses (ERVs) played a role during host–virus coevolution: from genomic domestication to functional innovation. Among others, it was argued (i) that long-terminal repeat (LTR) sequences can serve as a gene regulatory site and (ii) that endogenous retroviral envelope (env) genes can trigger innate immune responses [132].

### 8.10. The Story of Life

Life on earth began around 4–3.5 billion years ago with prebiotic chemistry, RNA, DNA, and proteins [133]. The prebiotic era was dominated by RNA, a versatile molecule that has been described in “The RNA world” [134]. Studies of the last two decades on the activities of RNA in our cells have revolutionized our understanding of the many roles played by this molecule. ncRNAs are ubiquitous and exert regulatory functions (see Section 7.10).

R Southwood described the story of life in less than 300 pages, summarizing the most important essentials [135]. Periods of relative calm were punctuated by five major extinctions: late Ordovician, late Devonian, end of Permian, late Triassic, and Cretaceous–Tertiary.

The first modern humans appeared in the Pleistocene. Meanwhile, more than 8 billion homo sapiens live on this planet and compete with other species for resources. This dominance and competition has a negative impact on biodiversity of plants and animals. In addition, homo sapiens is responsible for changing terrestrial landscapes and for pollution of environments by plastics and toxic chemicals.

About 500 million years ago, vertebrates developed and came up with increasingly sophisticated brain structures and with the sophisticated adaptive immunity system. About 1 million years ago, Homo erectus began with the controlled use of fire. The brain and immune system survived incredibly long through control mechanisms facilitating homeostasis. Neurons and lymphocytes learned to remain flexible to adapt to changing environmental conditions. The Nobel Laureate of Medicine from 2001, Sir Paul Nurse, has answered the question “What is life” by defining the following basic principles: (i) the capability to evolve by reproduction, including variability and natural selection; (ii) the capability to communicate with the environment in spite of the fact that life is based on cells that are demarcated from the environment by membranes; and (iii) the capability of information processing with a self-controlled metabolism. An example of his work is the spatiotemporal orchestration of mitosis by the enzyme cyclin-dependent kinase (CDK). The regulatory framework establishes core principles for control of the onset of cell division and highlights that the CDK control system operates within distinct regulatory domains in the nucleus and cytoplasm [136].

Intercellular communication, spatial orientation, learning, information processing and storage (memory) have been of great importance during evolution of vertebrates. The two parts of this review elucidate this with a focus on brain and the immune system. Without mutual interactions, protection and controlling between these two systems, 500 million years survival would not have been possible.

## Figures and Tables

**Figure 1 ijms-26-11567-f001:**
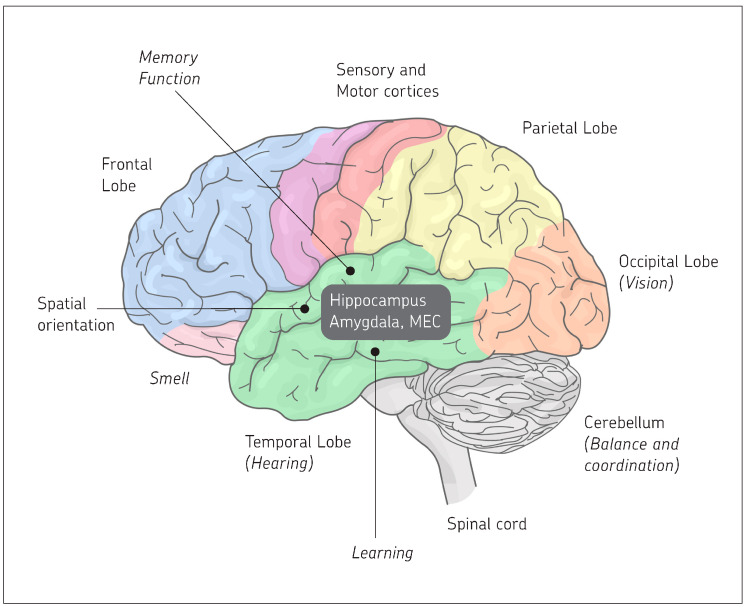
A simplified version of an external view of the human brain, showing lobes of the cerebrum and localization of major functions of the cerebrum and cerebellum. It integrates sites of spatial orientation, learning, and memory function deep in the temporal lobe. Other lobes are involved in top down and bottom up information processing. Processing sites in lobes of sensory inputs from vision, hearing, and smelling are indicated.

**Figure 2 ijms-26-11567-f002:**
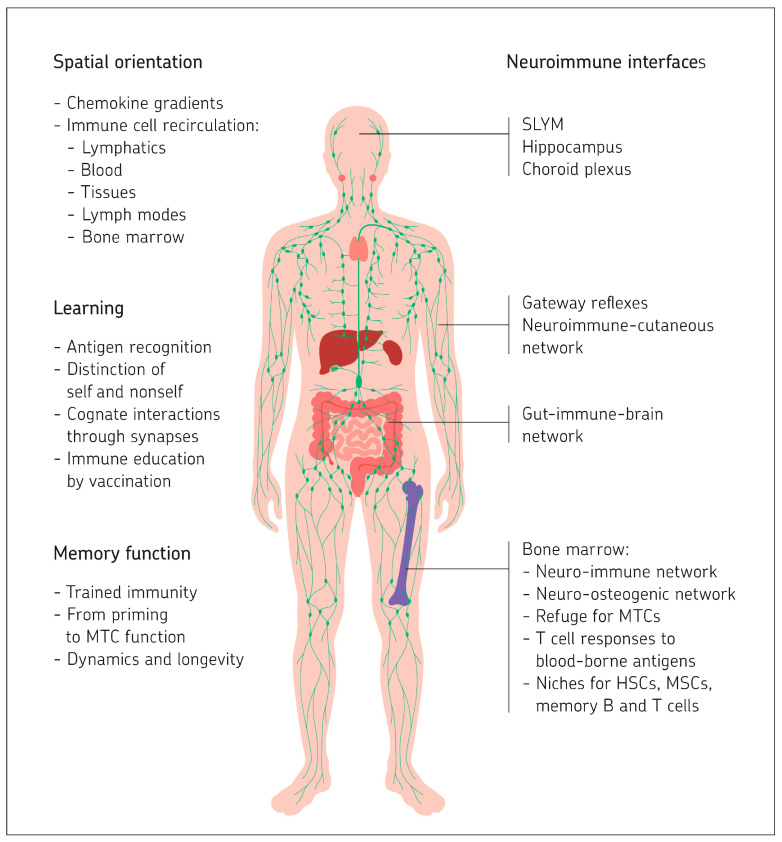
Sketch of the human body. The right side shows neuroimmune interfaces. These have been described in detail recently [2]. SLYM stands for subarachnoid lymphatic-like membrane, MTC for memory T cell, HSC for hematopoietic stem cell, and MSC for mesenchymal stem cell. The left side summarizes the main characteristics within the immune system of spatial orientation, learning, and memory function.

## Data Availability

No new data were created or analyzed in this study. Data sharing is not applicable to this article.

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
