# Peer review of "Brain and Immune System Part II—An Integrative View upon Spatial Orientation, Learning, and Memory Function"

_ijms, 2025, doi:10.3390/ijms262311567_

Round 1
Reviewer 1 Report
Comments and Suggestions for Authors
This review presents an interesting and highly interdisciplinary perspective, bridging neurobiology, immunology, and evolutionary biology.
To enhance its quality, the manuscript would benefit from further refinement in several points that are listed below:
- Depth of analysis: The work shows a good bibliographic effort and appears well-organized. However, some sections seem a bit vague due to their brevity, for example, -Immune system: Spatial orientation- or the section -Importance of the central immune system bone marrow- among others.
- A thorough proofreading is recommended to address minor typographical errors and punctuation issues, like the following examples:
- Line 8: There is a missing comma or period. Or an incorrect capital letter, maybe.
- Line 67: Consider change -accustical- for acoustical and add a point after “impressions”
- Suggestion: In Section 6, you could include information about the positive and negative selection processes of T lymphocytes in the thymus and B lymphocytes in the bone marrow. These processes represent a kind of initial training for lymphocytes and therefore shape the immune response through a sort of trial-and-error system within the immune system.
- Suggestion: In Figure 1, you have a great opportunity to illustrate the similarities mentioned in your work. You could, for example, use a lymphocyte that responds to a chemokine gradient, recognizes antigens through a synapse, or undergoes selection in the thymus as an example of learning, and then differentiates into a memory lymphocyte. You could place this schematic side by side with your simplified version of the brain, or arrange it as panel A) the brain and panel B) the lymphocyte. I believe this would enrich your work. This is only a suggestion—I do not intend to be intrusive regarding the author’s perspective.
- My PDF reader displays Figure 2 with very low resolution. It may be an issue with my computer, but I wanted to mention it in case the image quality needs to be improved at the source.
- Several of the main ideas in the article are supported by a single reference, which is acceptable; however, providing additional supporting evidence for the key arguments that conclude each paragraph would help reinforce their scientific validity and strengthen the overall foundation of the manuscript.
- Throughout the manuscript, there are several sentences and complete ideas that lack appropriate citations. Some examples; lines 141–143:
- For example, neurons in primary visual cortex might respond to a stimulus in a particular part of the visual field, a neuron in primary somatosensory cortex might respond to a tactile stimulus of a particular body part and the firing of a motor neuron might help to 144 direct limb movements in a specific direction.
- Lines 218-221:
- The term imprinting was coined to refer to cases of “programmed learning” during a critical period. Lorenz found that this period of partic-ular sensitivity to experience in geese is limited to a few hours soon after birth and that once imprinting occurs it is irreversible.
Author Response
Rev. 1
I thank Rev.1 for the supporting comments. Most of the suggestions
have been taken up for further refinement of the manuscript.
However, no changes have been made in Fig.1.
Reviewer 2 Report
Comments and Suggestions for Authors
The comparative approach between the brain and the immune system is highly interesting and intellectually stimulating.
The author successfully highlights functional parallels in spatial orientation, learning, and memory processes across the two systems, providing a cohesive and well-structured narrative.
However, the title “Brain and Immune System II: Spatial Orientation, Learning and Memory Function” could be reconsidered.
At first glance, it leads readers to expect a study focusing primarily on the specific roles of the brain and immune system in these functions, rather than a conceptual or comparative analysis.
A slightly revised title that better reflects the paper’s comparative and integrative nature would make the scope clearer to readers.
Aside from this point, the manuscript is well-organized, the arguments are coherent, and the topic is timely and engaging.
Author Response
Rev.2
I thank Rev.2 for the positive comments. The tiltle has been revised
according to the suggestion of Rev.2.
The revised manuscript includes a new chapter (8.4.) which deals with human-specific activities that distinguish homo sapiens from the animal kingdom.
All changes made in the revised version are highlighted in red.